# Subcuticular–Intracellular Hemibiotrophy of *Colletotrichum lupini* in *Lupinus mutabilis*

**DOI:** 10.3390/plants11223028

**Published:** 2022-11-09

**Authors:** Norberto Guilengue, Maria do Céu Silva, Pedro Talhinhas, João Neves-Martins, Andreia Loureiro

**Affiliations:** 1Instituto Superior de Agronomia, Universidade de Lisboa, 1349-017 Lisbon, Portugal; 2Agricultural Faculty, Agricultural Engineering Course, Instituto Superior Politécnico de Gaza, Lionde, Chókwè 1204, Mozambique; 3CIFC, Centro de Investigação das Ferrugens do Cafeeiro, Instituto Superior de Agronomia, Universidade de Lisboa, Pólo de Oeiras, 2784-505 Oeiras, Portugal; 4LEAF, Linking Landscape, Environment, Agriculture and Food, Associated Laboratory TERRA, Instituto Superior de Agronomia, Universidade de Lisboa, 1349-017 Lisbon, Portugal

**Keywords:** lupins anthracnose, fungal infection strategy, histology, ultrastructure

## Abstract

Anthracnose caused by *Colletotrichum lupini* is the most important disease affecting lupin cultivation worldwide. *Lupinus mutabilis* has been widely studied due to its high protein and oil content. However, it has proved to be sensitive to anthracnose, which limits the expansion of its cultivation. In this work, we seek to unveil the strategy that is used by *C. lupini* to infect and colonize *L. mutabilis* tissues using light and transmission electron microscopy (TEM). On petioles, pathogen penetration occurred from melanized appressoria, subcuticular intramural hyphae were seen 2 days after inoculation (dai), and the adjacent host cells remained intact. The switch to necrotrophy was observed 3 dai. At this time, the hyphae extended their colonization to the epidermal, cortex, and vascular cells. Wall degradation was more evident in the epidermal cells. TEM observations also revealed a loss of plasma membrane integrity and different levels of cytoplasm disorganization in the infected epidermal cells and in those of the first layers of the cortex. The disintegration of organelles occurred and was particularly visible in the chloroplasts. The necrotrophic phase culminated with the development of acervuli 6 dai. *C. lupini* used the same infection strategy on stems, but there was a delay in the penetration of host tissues and the appearance of the first symptoms.

## 1. Introduction

Lupins are legumes that belong to a large and diverse genus, *Lupinus*, comprising approximately 300 species, but only four species (*L. albus*, *L. angustifolius*, *L. luteus*, and *L. mutabilis*) present high agricultural importance [1,2,3]. Their seeds have one of the highest levels of protein among legumes, they offer potential health benefits, and they can contribute to the sustainability of cropping systems due to their net nitrogen input into the soil, which arises from their symbiotic interactions with rhizobia [1]. *L. mutabilis* (tarwi) is a species native to the Andes and, nowadays, is mainly cultivated throughout the Andean region, namely in Peru, Ecuador, and Bolivia. However, there is an increasing demand for protein crops that are suitable for Europe, and this species is a potential candidate [3]. *L. mutabilis* is characterized by the highest grain value of all cultivated lupins. It has a high protein content, an oil content that is similar to soybean, and is adapted to low input farming in temperate climates [4]. *L. mutabilis* cultivation is severely affected by anthracnose due to *Colletotrichum lupini* (Bondar) Damm, P.F. Cannon & Crous, which causes the twisting of stems, petioles, and pods and leads to the collapse of these tissues. In severe cases, progress of the disease causes plant death, compromising the crop [5].

The genus *Colletotrichum* was ranked in the top 10 fungal plant pathogens because of its broad host range, its ability to devastate essential crops, and its importance as a postharvest pathogen [6]. The pre-penetration phase is similar in all the pathogens of this genus, which usually begins with germination of the conidia and the formation of a melanized appressorium, a structure that is responsible for penetrating the host tissues [7,8,9]. After penetration, *Colletotrichum* species can use different strategies to infect and colonize host tissues, namely intracellular hemibiotrophy and subcuticular intramural necrotrophy, or even both [9,10,11,12]. The intracellular hemibiotrophic strategy is one of the most common and combines two phases: biotrophy and necrotrophy. After penetration, an infection vesicle and primary hyphae are formed. These structures do not kill the host’s invaded cells (biotrophic phase). Later in the infection process, necrotrophic secondary hyphae spread within host cells and kill the host tissue, culminating in the appearance of disease symptoms [9,13]. The extension of the biotrophic phase and the switch from biotrophy to necrotrophy can differ among *Colletotrichum* species depending on the host species, organ, and maturity stage [9]. The hemibiotrophic strategy has been widely studied and reported in different plants, including *Colletotrichum* spp. interactions, such as *Phaseolus vulgaris* with *C. lindemuthianum*, *Zea mays* with *C. graminícola* [7,14], *Sorghastrum nutans* with *C. caudatum* [15], *Tanacetum cinerariifolium* with *C. tanaceti* [16], *Arabidopsis thaliana* with *C. higginsianum* [17], and *Coffea arabica* with *Colletotrichum kahawae* [18,19]. In the subcuticular intramural necrotrophic strategy, the appressorium produces a penetration hypha that penetrates the host through the cuticle. The fungus grows under the cuticle within the periclinal and anticlinal walls of the epidermal cells without penetrating the protoplasts. Before entering the necrotrophic phase, the fungus causes swelling and dissolution of the epidermal cell walls and, afterwards, quickly grows both inter- and intracellularly throughout the tissue, causing cell disruption and death. There is no primary or secondary hyphae development [7,20]. This type of infection strategy has been observed in several interactions, such as *Gossypium hirsutum* with *C. truncatum* (as *C. capsici*) [7], *Capsicum annuum* and *Carica papaya* with *Colletotrichum truncatum* [21,22], and *Malus domestica* with *C. fructicola* [23]. In some particular cases, including the infection of almond fruits and leaves by *C. acutatum*, after subcuticular fungal penetration, there is a biotrophic phase where the fungus develops intracellularly before the necrotrophic hyphae start colonizing the host tissues, leading to cellular destruction and the development of symptoms [7,20,24]. Interestingly, the same *Colletotrichum* species can adopt different infection strategies in the same host depending on the organs to be colonized [24] and references therein. Thus, *Colletotrichum* species have dynamic interactions with their host and can implement different lifestyles. Understanding the pathogenic interactions of fungi with their hosts has important implications for disease control [7].

A recent study conducted on *Lupinus albus* infected with *C. lupini* that used transcriptomic and proteomic approaches revealed that the dynamics of symptoms, gene expression, and protein synthesis were similar to those of hemibiotrophic pathogens [25]. However, no studies have detailed the strategy that is used by *C. lupini* to infect and colonize lupin tissues. To overcome this lack of knowledge, in the present investigation a susceptible genotype of *L. mutabilis* was selected and inoculated with a *C. lupini* isolate, and the infection strategy was characterized using light and transmission electron microscopy. Once it was confirmed that the infection strategy applied by *C. lupini* was the same in both petioles and stems, we focused our thorough analysis on the behavior of the pathogen when colonizing lupin petioles. Petioles seem to be a more sensible tissue that displays symptoms earlier, as well as being a much easier organ to manage in histopathological procedures.

## 2. Results

### 2.1. Light and Transmission Electron Microscopy

#### 2.1.1. Conidial Germination and Appressorium Formation

The light microscopic analysis revealed that 1 day after inoculation (dai), about 50% of the conidia germinated and formed melanized appressoria on the surface of the petioles and stems. Around 30% of the spores germinated, but this was without appressoria formation. In the germinated conidia, close to 100% of the appressoria were sessile (Figure 1a). 

#### 2.1.2. Post-Penetration Stages

Petioles

In the petioles 1 dai, fungal penetration was evident by the appearance of internal light spots (Figure 1a) in more than 50% of melanized appressoria, as shown by the light microscopic observations. The internal light spot (ILS) corresponds to appressorium pore and penetration hypha development when viewed from above. Following cuticular penetration, the pathogen develops beneath the cuticle within the walls of epidermal cells and the apparent adjacent host cells appeared intact, as was observed at 2 dai (Figure 1b). No visible macroscopic symptoms were detected from 1–2 dai (Figure 1c).

However, at 3 dai, necrotic lesions first appeared (Figure 2a). Light microscopy showed that hyphal growth within the cuticle was followed by the penetration of epidermal cells and further inter- and intracellular colonization of the cortex and less frequently of the vascular cells (Figure 2b,c). 

No distinct primary and secondary hyphae seem to have developed. The dissolution of cell walls was observed, particularly in the epidermal cells (Figure 2b,c). Disorganization of the cytoplasmic contents of some of the host’s infected cells was also detected (Figure 2b,c). At this time point, transmission electron microscopy demonstrated the intense degradation of epidermal cell walls and the different levels of plasma membrane and cytoplasm disorganization: (i) discontinuities appeared in the plasma membrane, which retracted away from the host cell wall; and (ii) cells showed fragments of the host plasma membrane and other remaining cellular debris. In the epidermal and cortex cells, it was frequently observed that chloroplast membranes were losing their integrity (Figure 3). 

At 4 dai, the intense degradation of the walls of epidermal cells and those of the first layers of the cortex were visible by light microscopy. The cytoplasm of these cells collapsed or was disrupted, with only fragments remaining (Figure 4a,b).

At 6 dai, acervuli had been formed. Acervuli were initiated by the formation of a dense hyphal stroma in the epidermis that gave rise to conidiophores, which disrupted the cuticle with the subsequent release of conidia and the appearance of dark sunken lesions (Figure 5a,b).

Stems

The strategy applied by *C. lupini* to infect stem tissues was the same as that which was applied to the petioles. The initial subcuticular–intramural penetration and biotrophic intracellular growth were followed by necrotrophic fungal growth and the development of symptoms (Figure 6). However, there was a delay in the fungal penetration and colonization of stem tissues. The fungal penetration was not observed until 3 dai (Figure 6a), and the first symptoms only appeared from 6 dai.

A diagram of the infection process that is used by *Colletotrichum lupini* to colonize *Lupinus mutabillis* petioles and stems is represented in Figure 7.

## 3. Discussion

To the best of our knowledge, this is the first histological and ultrastructural study concerning the infection process of *Colletotrichum lupini* in *Lupinus* tissues. As in other *Colletotrichum*–plant pathosystems [7,18,20,21,22], the early stages of fungal development involved conidia germination and the differentiation of melanized appressoria. Highly specialized, melanized appressoria have extremely high turgor pressure, which is crucial for cuticle penetration [26]. At 1 dai, the cuticles of the petioles were already penetrated, which was confirmed by the presence of appressorial internal light spots. According to Diéguez-Uribeondo et al. (2003) [27], the internal light spot corresponds to the penetration pore from which the infection peg emerges, and host colonization begins. These are considered indicators of successful penetration. After cuticle penetration, the fungus grew beneath the cuticle and the epidermal cell wall, which is similar to what has been described in interactions between *Capsicum annuum*, *Carica papaya* and *Colletotrichum truncatum* [21,22], *Malus domestica* and *C. fructicola* [23], and *Fragaria × ananassa* and *C. acutatum* [28]. The host epidermal cells that were adjacent to these subcuticular intramural hyphae appeared intact, suggesting the existence of a biotrophic fungal growth phase. Identical biotrophic development was also observed during the subcuticular intramural colonization of apple leaves by *C. fructicola* [23]. In *L. mutabilis* petioles, hyphal growth within the cuticle was followed by the penetration of epidermal cells and by further inter- and intracellular colonization of the cortex and vascular cells. The ultrastructural modifications observed 3 dai suggested that this time point corresponds to the switch for necrotrophy. Besides the intense degradation of epidermal cell walls, different levels of plasma membrane and cytoplasm disorganization were reported. Some cells showed discontinuities in the plasma membrane which retracted away from the host cell wall, other cells exhibited fragments of the host plasma membrane and other remaining cellular debris, and in other cells, it was possible to observe the loss of chloroplast membrane integrity. Analogous cellular alterations were described in the interactions between coffee and *C. kahawae* [18] and between apple and *C. fructicola* [23] when the necrotrophic phase was established. The growth of *C. lupini* in *L. mutabilis* during the necrotrophic phase was associated with intense cell wall degradation and the death of the host’s protoplasts, and culminated in the production of acervuli. As observed in the *L. mutabilis* petioles, *C. lupini* used the subcuticular intracellular hemibiotrophic strategy to infect and colonize the stems, as was revealed by the histological studies. Most *Colletotrichum* species are hemibiotrophic [9]. Interestingly, the transcriptomic and proteomic data obtained by Dubrulle et al. (2020) [25] also support the existence of a hemibiotrophic phase in *L. albus* and *C. lupini* interactions.

Lupin anthracnose is notorious for affecting stems. Since lupins bloom apically, the effect of stem infection hampers fruit set and, therefore, can be disastrous. Lupin anthracnose symptoms are peculiar due to the twisting of the affected organs (petioles, stems, and pods), a situation seldom found in anthracnose pathosystems (e.g., onion twister disease or celery anthracnose) [5]. The collapse of host cells in the infected area, as documented in the present study, could explain the twisting of lupin organs. Although *L. mutabilis* is generally regarded as susceptible, differences in disease responses have been noted [29] and seem to be influenced by anthocyanin pigmentation [30]. In the present work, a highly susceptible genotype was chosen, enabling the detailed characterization of infection mechanisms. We believe that this information in *Lupinus* will allow plant breeders to develop better control strategies against this disease and will enable better analysis of these interactions with less susceptible genotypes.

## 4. Materials and Methods

### 4.1. Biological Material

For the present work, a *Colletotrichum lupini* isolate (RB221, isolated from *L. albus* in France [29,31]) was used. *L. mutabilis* LM231 accession was selected as vegetal material due to its high susceptibility to anthracnose [29]. Plant and fungus growth conditions were the same as those described by Guilengue et al. (2020) [29]. 

### 4.2. Inoculation and Incubation

Whole plants (3–4 weeks old, containing 6–8 leaves) were placed horizontally on plastic trays on a wet nylon sponge and then the stem and petioles were inoculated with a 10μL drop of a conidia suspension (2 × 10^6^ spores/mL). In order to ensure the adhesion of spores to the plant, the spore suspension contained 1% gelatin [29]. The covered trays were incubated for the first 24 h in a dark moist chamber at 22 °C and were then kept for a photoperiod of 12 h at the same temperature.

The plants that were used as controls were treated exactly like the inoculated ones, but instead of conidial suspension, distilled water was used.

### 4.3. Sampling and Microscopy

Conidial germination and appressoria differentiation were evaluated on the petioles and stem pieces (≈5 cm^2^) 1 day after inoculation (dai), as previously described [32]. The samples were painted with transparent nail polish on the inoculated surface in order to recreate a tissue surface replica [33]. Once dried, the nail polish was removed, stained, and mounted in lactophenol cotton blue. For each experiment, a minimum of six microscope fields, each containing 100 conidia and/or differentiated appressoria on the surface of the tissues, were used. 

To visualize the progression of the *C. lupini* infection under a bright-field, the petiole and stem samples were collected (at 1, 2, 3, 4, 6, and 9 dai) and were fixed using a technique adapted from Fernandes et al. (2021) [34]. The sample pieces were immersed in formaldehyde (37 to 38%)/glacial acetic acid/ethanol (70%) in a 1:1:18 proportion for a minimum period of 24 h. Before sectioning, the tissues were rehydrated with distilled water for 10 min. Sectioning (20–25 µm thick) was performed using a freezing microtome (Leica CM1850; Zurich, Switzerland).

To analyze fungal post penetration stages, the cross-sections were stained and mounted in cotton blue lactophenol [18]. In order to optimize the observation of the fungal infection process, strips from healthy and infected tissues (at the same time-points used to collect fresh tissues: 1, 2, 3, 4, 6, and 9 dai) were prepared according to a technique previously described [32]. The tissues were cut and fixed in a 2.5% (*v*/*v*) solution of glutaraldehyde in 0.1M sodium cacodylate buffer at pH 7.1. After 2 h, the tissues were washed (3 × 20 min) in sodium cacodylate buffer and were post-fixed with 1% osmium tetroxide in the same buffer for 2 h. The tissues were then washed in distilled water for 10 min. Subsequently, the tissues were dehydrated in a graded ethanol series (10–100% at 10% increments), embedded in Spurr’s resin (TAAB), and polymerized overnight at 70 °C. As it was difficult to impregnate the plant material with resin, alterations to the described technique were introduced. The dehydration periods in ethanol were increased (20 min at each dehydration point up to 90% and two rinses were performed with 100% ethanol for 1 h each). The process of embedding the tissues in Spurr’s resin was also more gradual (proportion resin:ethanol: 1:3, 1:2, 1:1, 2:1, and 3:1 for 12 h each), and was performed at 4 °C with agitation (ca. 10 rpm). Semi-thin sections (2 µm) that were made with an ultramicrotome Leica (Leica Ultracut R. Zurich, Switzerland) were stained with 0.5% aqueous toluidine blue O solution and were observed using light microscopy. Observations were made using a light microscope Leica DM-2500.

In order to better elucidate the infection strategy of *C. lupini,* polymerized blocks of healthy and infected material collected at 3 dai (corresponding to the appearance of the first necrosis) were used for transmission electron microscopic observations. Ultrathin sections (80–90 nm) of polymerized blocks were made with a diamond knife (mod. DIATOME ultra 458; DIATOME, Hatfield, PA, USA) using an ultramicrotome (Leica Ultracut R. Zurich, Switzerland). The sections were then collected on Formvar-coated nickel grids (200 mesh), stained with uranyl acetate and Reynold’s lead citrate, and were examined using a 120 kV Transmission Electron Microscope (Hitachi H-7650. Tokyo, Japan) equipped with a XR41M mid mount AMT digital camera for data acquisition.

## Figures and Tables

**Figure 1 plants-11-03028-f001:**
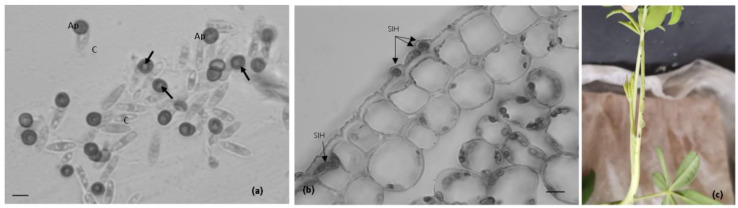
The *Colletotrichum lupini* infection process on petioles of *Lupinus mutabilis* and the corresponding macroscopic symptoms. (**a**,**b**). Light microscopic observations, cotton blue lactophenol staining. (**a**). Conidia (C) germination and appressoria (Ap) formation, 1 day after inoculation (dai). Note the appressoria light spots (arrows) on melanized appressoria. (**b**). Subcuticular intramural hyphae (SIH) developing between the cuticle and epidermal cell walls 2 dai. (**c**). Petioles without disease symptoms 2 dai. Scale bar = 10 µm.

**Figure 2 plants-11-03028-f002:**
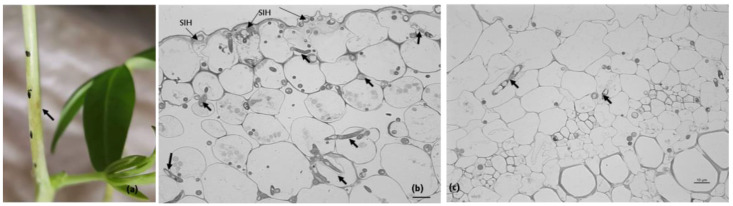
Macroscopic symptoms and the corresponding stage of the *Colletotrichum lupini* infection process on petioles of *Lupinus mutabilis* 3 days after inoculation. (**a**). Beginning of disease symptoms (arrow). (**b**,**c**). Light microscopic observations, toluidine blue staining. (**b**). Infection sites showing subcuticular intramural hyphae (SIH) and hyphae developing intra- and intercellularly (arrows) in epidermal and cortex host cells. Note the degradation of some epidermal cell walls. (**c**). Intra- and intercellular fungal growth (arrows) in cortex cells and vascular cells. Scale bar = 10 µm.

**Figure 3 plants-11-03028-f003:**
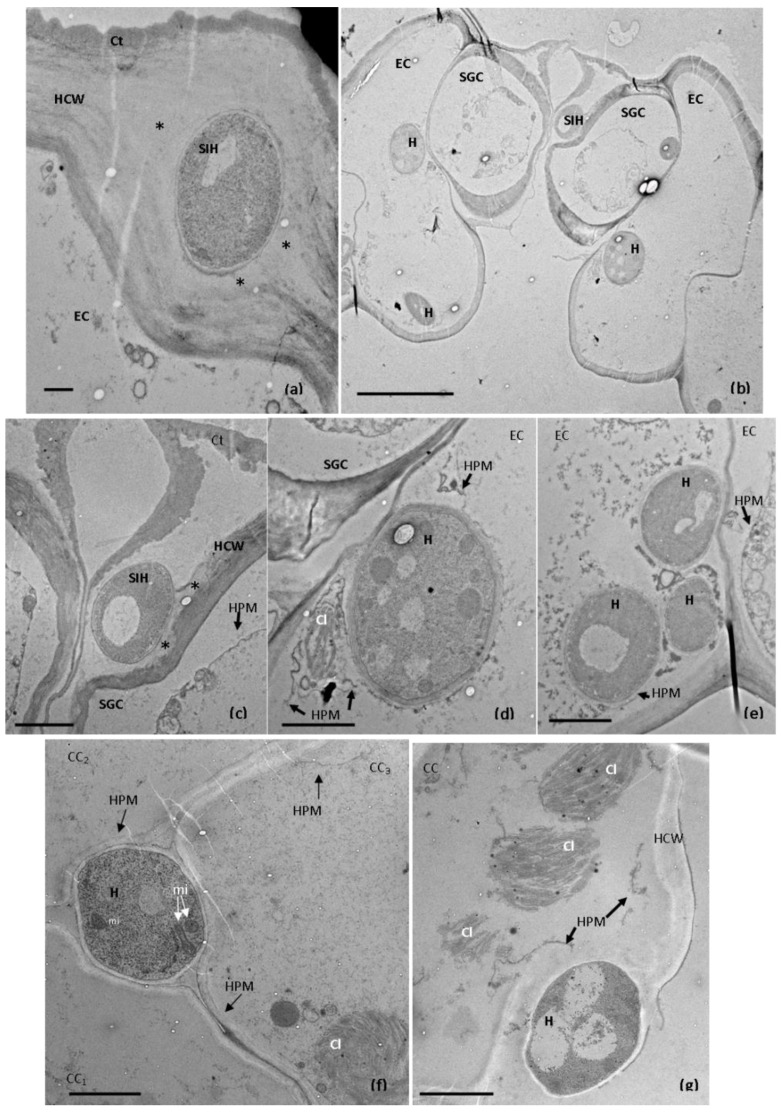
Electron micrographs showing colonization of *Lupinus mutabilis* petioles by *Colletotrichum lupini* 3 days after the inoculation. (**a**). A subcuticular intramural hypha (SIH) expanded between the cuticle (Ct) and the epidermal host cell wall (HCW). Degradation of the epidermal cell (EC) wall (asterisks) adjacent to the hypha is visible. Scale bar = 500 nm. (**b**). A subcuticular intramural hypha (SIH) near the wall of a stomatal guard cell (SGC) and intracellular hyphae (H) in epidermal adjacent cells (EC). Scale bar = 10 µm. (**c**). Enlargement of “b”, showing the degradation/dissolution of the wall (asterisks) of the stomatal guard cell (SGC) near the subcuticular intramural hypha (SIH). The plasma membrane of the stomatal guard cell apparently remained intact. Scale bar = 2 µm. HCW = host cell wall. (**d**). Enlargment of “b”, showing a hypha (H) within an epidermal cell (EC). The host plasma membrane (HPM) retracted away from the host cell wall, and it is possible to see its breakdown. The chloroplast (Cl) membranes are losing their integrity. Scale bar = 2 µm. (**e**). Hyphae (H) within an epidermal cell (EC), showing the breakdown of the host plasma membrane (HPM) and disorganization of its cytoplasmic content. Scale bar = 2 µm. (**f**). Intercellular hypha (H) between cortex cells (CC1–CC3), which present different levels of disorganization of their cytoplasmic contents. In CC1 and CC2 cellular debris is observed. The plasma membrane is only visible in CC2. In CC3, discontinuities appeared in the plasma membrane and its retraction away from the host cell wall was also visible. This cell (CC3) also shows one chloroplast (Cl) with disorganized membranes. Mi = mitochondria. Scale bar = 2 µm. (**g**). Hypha within the wall of a cortex cell (CC), showing discontinuities in the plasma membrane (HPM) and disorganization of the chloroplast (Cl) membranes. Scale bar = 2 µm.

**Figure 4 plants-11-03028-f004:**
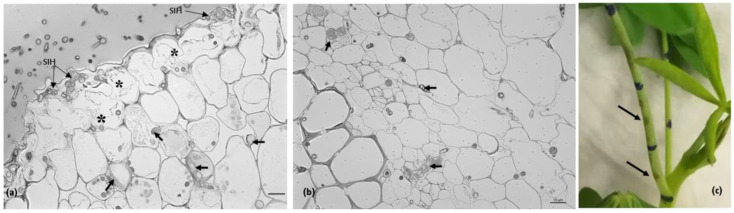
The *Colletotrichum lupini* infection process on petioles of *Lupinus mutabilis* and the corresponding macroscopic symptoms 4 days after inoculation. (**a**,**b**). Light microscopic observations, toluidine blue staining. (**a**). Infection sites showing subcuticular intramural hyphae (SIH) and hyphae developing intra- and intercellularly (arrows) in epidermal and cortex host cells. Note the intense degradation of the epidermal cell walls and those of the first layers of the cortex, and also the collapse or disruption of the cytoplasm of some of these infected cells (*). (**b**). Intra- and intercellular fungal growth (arrows) in cortex cells and vascular cells. Plant cell degradation is not so intense as observed in the epidermal cells and those of the first layers of the cortex cells (see Figure 2b). (**c**). Petioles exhibiting disease symptoms (arrows). Scale bar = 10 µm.

**Figure 5 plants-11-03028-f005:**
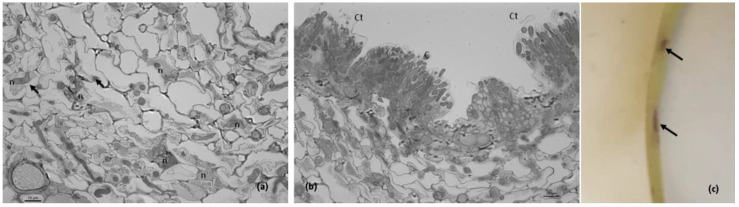
The *Colletotrichum lupini* infection process on petioles of *Lupinus mutabilis* and the corresponding macroscopic symptoms 6 days after inoculation. (**a**,**b**). Light microscopic observations, toluidine blue staining. (**a**). Dense fungal colonization of petiole cells, including the vascular system. Note the necrotized host cells (n) and their intense degradation and deformation. (**b**). Acervuli production, disruption of the petiole cuticle (Ct), and the release of conidia (C). (**c**). Petioles with dark sunken lesions (arrows). Scale bar = 10 µm.

**Figure 6 plants-11-03028-f006:**
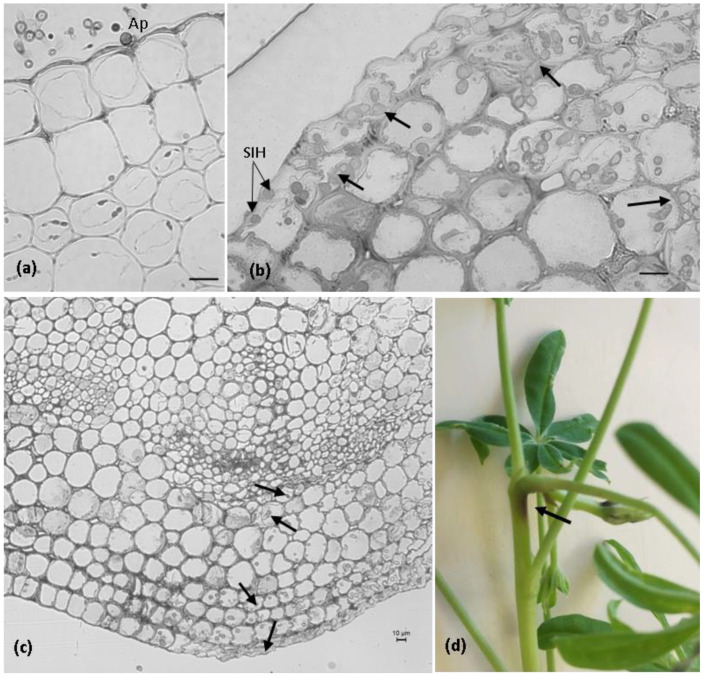
The *Colletotrichum lupini* infection process on the stems of *Lupinus mutabilis* and the macroscopic symptoms. (**a**–**c**). Light microscopic observation, toluidine blue staining. (**a**). Infection site showing a melanized appressorium (Ap) 3 days after inoculation (dai). (**b**). Infection sites showing subcuticular intramural hyphae (SIH) and hyphae developing intra- and intercellularly (arrows) in epidermal and cortex host cells 6 dai. (**c**). Fungal colonization extended from the cuticle and epidermal cells to the vascular system (arrows). Note the intense degradation and deformation of epidermal and cortex plant cells 9 dai. (**d**) Stem with dark sunken lesions (arrow) 9 dai. Scale bar = 10 µm.

**Figure 7 plants-11-03028-f007:**
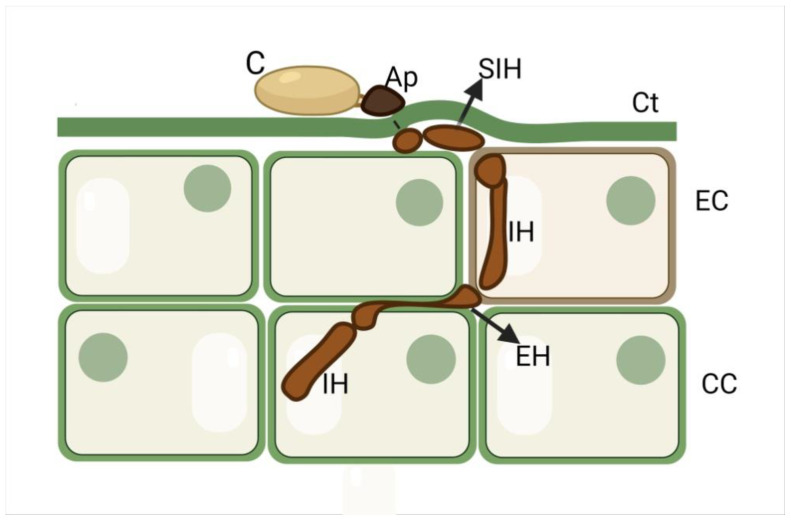
Generalized diagram of the infection strategy used by *Colletotrichum lupini* in *Lupinus mutabillis* petioles and stems: subcuticular–intracellular hemibiotrophy. Conidium (C) germinates to form melanized appressorium (Ap), from which penetration hypha develop. The host cuticle (Ct) is penetrated and fungal hyphae spread in a subcuticular fashion within the walls of the host epidermal cells (ECs). This intramural development is associated with the dissolution of the host cell walls, but the adjacent host cells appeared intact (biotrophy). The subsequent steps of the infection involved the penetration of epidermal cells and further inter- and intracellular colonization of cortex (CC) and vascular cells. The fungus progressively killed the infected host cells and dissolved the cell walls (necrotrophy). IH = intracellular hypha; EH = extracellular hypha. The diagram is not to scale. Created with Biorender.com.

## Data Availability

Not applicable.

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
