# Peer review of "Subcuticular–Intracellular Hemibiotrophy of Colletotrichum lupini in Lupinus mutabilis"

_plants, 2022, doi:10.3390/plants11223028_

Round 1

Reviewer 1 Report

The study “Subcuticular-intracellular hemibiotrophy of Colletotrichum lupini in Lupinus mutabilis” by Guilengue et al. is using light- and electron microscopy to investigate the nature of the infection process of C. lupini on L. mutabilis. This specific interaction has not been analyzed in this way so far. The study is conducted soundly and the manuscript is well written. I recommend the manuscript for publication after minor revisions. I would like to encourage the authors to further improve the study by adding a schematic drawing of the infection process based on their findings (like for example shown in https://doi.org/10.1007/s13225-021-00487-5 , Fig. 4.). This is not a compulsory request, but could help to increase citation rate, since these schemes are useful for e. g. lectures and presentations.

- Line 106: the term “internal light spots” is used for the first time. Is this structure corresponding to the appressorial cone (cf. respective literature on appressoria formation) and the penetration hypha viewed from above? If so, please also state this here.

- Figure 3: in the legend “HCW” and “EC” are not mentioned. Please add the abbreviations to the legend. Please consider using more equal font sizes for the markings in the figure.

- Please elaborate in the discussion if you think, that the same infection process could also be observed on L. angustifolius, which is currently of higher agronomical importance

Author Response

We are pleased to submit the revised version of the manuscript. All the corrections performed in the document were listed as track changes. We would like to thanks the reviewers for the thoughtful comments and constructive suggestions, which allowed us to improve the quality of this manuscript.

We have addressed all of the reviewers concerns and suggestions and the manuscript was revised accordingly.

Please find below a point by point response to your questions:

  • Line 106: the term “internal light spots” is used for the first time. Is this structure corresponding to the appressorial cone (cf. respective literature on appressoria formation) and the penetration hypha viewed from above? If so, please also state this here.

         Sentence re-written

  • Figure 3: in the legend “HCW” and “EC” are not mentioned. Please add the abbreviations to the legend. Please consider using more equal font sizes for the markings in the figure

          Correction performed

  • Please elaborate in the discussion if you think, that the same infection process could also be observed on L. angustifolius, which is currently of higher agronomical importance.

Unlike most species of Colletotrichum, C. lupini is specialized in Lupinus spp. and there are evidences that both entities have co-evolved in South America before spreading to other regions. Whereas Lupinus has build up a large morphological, citogenetic and geographical diversity, C. lupini has remained nearly clonal. Macroscopically anthracnose symptoms are very similar across lupins, including the typical twisting of stems and petioles, causing damages that become of economic importance to the main lupin crops, white-lupin, yellow-lupin, narrow-leaved-lupin and tarwi. It is possible that the infection process here described for C. lupini on tarwi is similar on other lupin crops, but such is no more than a working hypothesis that must be addressed by research on other lupin species.

  •  I would like to encourage the authors to further improve the study by adding a schematic drawing of the infection process based on their findings (like for example shown in https://doi.org/10.1007/s13225-021-00487-5 , Fig. 4.). This is not a compulsory request, but could help to increase citation rate, since these schemes are useful for e. g. lectures and presentations.

We have made a scheme according with your suggestion

Reviewer 2 Report

Dear Authors,

Please improve the clarity of selected sentences and check my suggestions in the manuscript text.

In my opinion the infection strategy could be presented in 2 different species (or cultivars) of lupine, to compare the highly susceptible cultivars with the less susceptible. This knowledge would be helpful for the lupine growers in the crop protection.

Author Response

We are pleased to submit the revised version of the manuscript. All the corrections performed in the document were listed as track changes. We would like to thanks the reviewers for the thoughtful comments and constructive suggestions, which allowed us to improve the quality of this manuscript.

We have addressed all of the reviewers concerns and suggestions and the manuscript was revised accordingly.

  • Please improve the clarity of selected sentences and check my suggestions in the manuscript text.

Changes were made as listed below:

Line 18

Correction performed

Line 36

Sentence re-written

Line 41

Sentence re-written

Line 129

Correction performed

Line 141

Correction performed

 Line  169

96hai replaced by 4 dai

Line 252

Correction performed

Line 264

Correction performed

Line 270

Correction performed

Line 274

Sentence re-written

Line 276

Correction performed

Line 277

Correction performed

Line 284

Correction performed

Line 288

Sentence re-written

Line 292

Sentence re-written

Line 307

Correction performed

Line 334

Correction performed

  • In my opinion the infection strategy could be presented in 2 different species (or cultivars) of lupine, to compare the highly susceptible cultivars with the less susceptible. This knowledge would be helpful for the lupine growers in the crop protection.

Thank you very much for your suggestion. As a first approach our main focus was to study the infection mechanisms per si in a susceptible variety and to optimize the techniques to be used. From here we believe that we have pave the way to perform new and interesting studies that can include others varieties or species and their comparison. We believe that this will be done in a near future.